# Learning Temporal–Spatial Contextual Adaptation for Three-Dimensional Human Pose Estimation

**DOI:** 10.3390/s24134422

**Published:** 2024-07-08

**Authors:** Hexin Wang, Wei Quan, Runjing Zhao, Miaomiao Zhang, Na Jiang

**Affiliations:** College of Information Engineering, Capital Normal University, Beijing 100048, China; 2221002098@cnu.edu.cn (H.W.); 2221002050@cnu.edu.cn (W.Q.); 2231002030@cnu.edu.cn (R.Z.); 6830@cnu.edu.cn (M.Z.)

**Keywords:** 3D human pose estimation, dual-adaptive spatial-temporal model, one-more supervised training, batch variance loss

## Abstract

Three-dimensional human pose estimation focuses on generating 3D pose sequences from 2D videos. It has enormous potential in the fields of human–robot interaction, remote sensing, virtual reality, and computer vision. Existing excellent methods primarily focus on exploring spatial or temporal encoding to achieve 3D pose inference. However, various architectures exploit the independent effects of spatial and temporal cues on 3D pose estimation, while neglecting the spatial–temporal synergistic influence. To address this issue, this paper proposes a novel 3D pose estimation method with a dual-adaptive spatial–temporal former (DASTFormer) and additional supervised training. The DASTFormer contains attention-adaptive (AtA) and pure-adaptive (PuA) modes, which will enhance pose inference from 2D to 3D by adaptively learning spatial–temporal effects, considering both their cooperative and independent influences. In addition, an additional supervised training with batch variance loss is proposed in this work. Different from common training strategy, a two-round parameter update is conducted on the same batch data. Not only can it better explore the potential relationship between spatial–temporal encoding and 3D poses, but it can also alleviate the batch size limitations imposed by graphics cards on transformer-based frameworks. Extensive experimental results show that the proposed method significantly outperforms most state-of-the-art approaches on Human3.6 and HumanEVA datasets.

## 1. Introduction

Three-dimensional human pose estimation is a rapidly developing task at the intersection of computer vision [1,2,3], human–robot interaction [4,5,6], and sensors [7,8]. It enables reconstruction of the 3D structure of the human body from 2D images or videos, providing a more comprehensive and detailed understanding of human movement and behavior [9,10]. The advancements in 3D human pose estimation are driven by the remarkable progress in deep learning techniques and computer vision methods. The combination of these technologies has enabled more accurate segmentation and localization of body parts [11,12,13], as well as the reconstruction of their 3D spatial configuration [14,15].

The initial 3D human pose estimation based on deep learning takes single 2D image/pose as input [16,17,18]. Due to the lack of depth information, the estimated pose may appear accurate on the imaging plane, but there is a significant deviation in the world coordinates. Taking Figure 1 as an example, the second line displays three failed results from videoposed3D [16]. videoposed3D is a representative work for 3D pose estimation using a single image as input. However, when facing problems such as occlusion, complex actions, and cluttered backgrounds, accurate pose estimation is still not possible. The main reason behind this is the lack of depth clues. It is difficult to obtain accurate depth information from monitoring data or remote sources in real scenes. The posture changes expressed by continuous 2D sequences can alleviate this problem. Therefore, video-based 3D pose estimation has emerged and gradually become a research hotspot.

Related algorithms generally aim to encode the spatial locations and temporal variations of the pose sequence [20,21]. In some early works, convolution was often used in the temporal or spatial encoder [16,22]. On this basis, graph convolution is introduced to utilize the spatial correlation between different keypoints [10,23]. Recently, in response to the difficulty of inferring invisible joints, Yang et al. [24] proposed a weak supervision framework for 3D human pose estimation from a single image, which can be applied not only to 2D-to-3D pose pairs, but also to 2D individual annotations. Moreover, some methods [25,26] combine global adaptation and local generalization, which is a simple and effective unsupervised domain adaptation framework for 3D human pose estimation. With the success of transformer in other vision tasks, the transformer block has become a core component of current network architecture design [27,28,29]. For instance, MAED [30] designed a spatial–temporal encoder based on a transformer block to independently extract temporal and spatial features for 3D pose estimation. On this basis, MotionBert [31] proposed a two-branch DSTformer. It contains temporal-to-spatial and spatial-to-temporal models, which can realize pose inference from 2D to 3D through direct parallel fusion. Although these methods have promoted the development of 3D pose estimation, they have neglected the spatial–temporal synergistic effects. This leads to the estimated 3D pose sequence being prone to missing details or lagging changes.

To compensate for this deficiency, we propose in this paper a dual-adaptive spatial–temporal former (DASTFormer). It achieves spatial–temporal collaborative encoding through attention- adaptive (AtA) and pure adaptive (PuA) modes. The network’s three-branch design serves two main functions: it reduces the impact of the spatial transformer block (STB) on the temporal transformer block (TTB) when they are arranged sequentially, and it addresses the issue of overlooking the overall spatial and temporal consistency of joints that can occur with a parallel arrangement. Furthermore, regarding the feature fusion module, PuA ensures global stability, while AtA ensures local accuracy. This design can enhance the different effects of temporal and spatial encoding on each keypoint, allowing a more accurate estimation of the 3D pose. Furthermore, a novel one-more supervised training strategy with batch variance loss (BVLoss) is designed to seek the global optimum. Two-round parameter updates are performed on the same batch. And the second forward propagation result is required to be better than the first one. Through this strategy, the proposed method will deeply explore the potential association between spatial–temporal encoding and 3D pose estimation. Meanwhile, it can also alleviate the batch size limitation of graphics cards on DASTFormer. To verify the effectiveness of our proposed approach, we conduct on a series of experiments on popular Human3.6M and HumanEVA datasets. The results demonstrate that our proposed approach achieves outstanding performance on MPJPE (mean per joint position error) and P-MPJPE (MPJPE after rigid alignment by pose-processing).

Our contributions are summarized as follows:Designed a dual-adaptive spatial–temporal former encoder, which contains a pure adaptive mode and an attention adaptive mode. It can enhance the different effects of temporal and spatial encoding on each keypoint to improve 3D pose estimation.Designed a one-more supervised training strategy with batch variance loss, which conducts a two-round parameter update for the same batch data. It can effectively explore potential 3D location cues from 2D inputs and alleviate the batch-size limitation of the GPU (graphics processing unit).State-of-the-art performance was achieved on Human3.6M and HumanEVA datasets. Apart from that, the recovery effect is better in some complex postures and in wild images.

## 2. Related Work

Three-dimensional human pose estimation is a task that aims to estimate the spatial location of human body key points from images or videos. This task has numerous applications in areas such as animation, remote sensing, virtual reality, and healthcare. Over the years, numerous algorithms and techniques have been proposed to address this problem. In this section, we discuss and analyze existing work from three aspects: input form, spatial–temporal cue, and training strategy.

### 2.1. Image and Video Inputs

In the initial studies, images served as the primary input format for 3D human pose estimation [32,33]. Zhou et al. [34] employed a weakly supervised approach, incorporating a mix of 2D and 3D labels for network training. However, this approach did not exploit sequential frames to address challenges such as occlusions, leading to unreliable performance in complex scenarios, particularly with occlusions. Building on the insights of Pavllo et al. [16], subsequent research began to consider video input by integrating temporal convolutional networks (TCN). The capacity of TCN to handle sequential data through temporal convolution proved to be advantageous. However, this method has not thoroughly explored the integration of temporal and spatial information, nor has it separately modeled spatiotemporal information. Additionally, it has not considered the positional relationships between human joints, leading to an excessive reliance on the accuracy of 2D keypoint detection in the network. Hossain and Little [35] took a step further by leveraging temporal information from a sequence of 2D joint positions to estimate a series of 3D poses. This work relied on a purpose-designed sequence-to-sequence network, incorporating long short-term memory (LSTM) units with normalized layers and temporal smoothness constraints for training [36,37]. Animepose [38] utilized scene LSTM to predict one-step actions of a person. Specifically, the initial step involves forecasting the action in the preceding frame, followed by the estimation of joint positions in subsequent frames based on the key point sequence. Lee et al. [39] introduced a novel architecture built on LSTM, with the aim of learning intrinsic representations to reconstruct 3D depth from centroids to key points. Due to the absence of depth information in 2D images, directly obtaining 3D pose from 2D images is highly unreliable. Hence, this paper focuses on video-based 3D human pose estimation. Utilizing video input offers continuous frames, allowing for improved joint localization through inter-frame complementarity. However, most methods, such as those that employ LSTM to extract temporal cues [38,39], often neglect the importance of spatial information, potentially leading to a lack of details in the estimated 3D sequences.

### 2.2. Spatial and Temporal Clues

In addition to network structures such as CNN and RNN, the adaptation of transformers, which has seen tremendous success in the NLP domain, has also produced significant advancements in computer vision. ViT [40] was the pioneer in applying Transformers for classification in computer vision. Yang et al. [41] introduced a network that leverages transformers to extract 2D poses, showcasing the versatility and effectiveness of this approach in the field of pose estimation. Zheng et al. [20] developed a transformer network based on ViT for 3D human pose estimation.

In the past two years, an increasing number of methods have been based on spatial–temporal transformers for this task, highlighting their growing popularity and effectiveness in the field of 3D human pose estimation [20,21,28,42,43]. Shen et al. [44] adopted a mask pose and shape estimation strategy to introduce a global transformer for long-term modeling. This strategy randomly masks features of several frames to stimulate the global transformer to learn more interframe dependencies. The local transformer is responsible for utilizing local details on the human mesh and interacting with the global transformer through the use of cross-attention. However, Zheng et al. [20] requires a fixed order of spatial and temporal encoders and only reconstructs the central frame of a video. In addition, recent work has focused on optimizing the transformer structure, given the substantial computational demands and complexity involved. Einfalt et al. [45] proposes a transformer-based pose-boosting scheme that can operate on time-sparse 2D pose sequences, but still produces time-intensive 3D pose estimation. It also demonstrates how mask labeling modeling can be used for temporal upsampling within transformer blocks, greatly reducing the overall computational complexity. Li et al. [46] proposed an improved architecture based on transformers, which simply promotes long-sequence 2D joint positions to a single 3D pose, effectively aggregating remote information into a single vector representation in a hierarchical global and local manner, significantly reducing the computational cost. Even if these methods all reduce the amount of computation, due to insufficient utilization of global information and using only the consistency of adjacent frames to solve, performance may be affected.

Overall, many of the above methods use temporal and spatial information, but they do not effectively integrate the temporal and spatial information, which can lead to the network leaning toward one side and failing to achieve the role of global perception, leading to inefficient capture of global context by the network. This results in limited performance in tasks that demand a comprehensive understanding of spatial–temporal relationships.

### 2.3. Diverse Training Strategies

Three-dimensional human pose estimation includes end-to-end recovery from the original image [10,32,47,48] and using 2D joint points extracted from the original image, recovering the 3D pose through joint point mapping between 2D and 3D [22,49,50,51,52,53]. However, recovering 3D human pose directly from the original image requires a higher computational cost. And moreover, the background and noise present in the original image often have a counterproductive effect on recovering 3D poses, significantly enhancing both the complexity and computational load of 3D human body recovery. Recently, methods such as CPN [13], AlphaPose [54], and HRNet [55] have become commonly used 2D joint detection algorithms. For example, HRNet [55] gradually increases the number of stages by maintaining high-resolution throughout the process, progressively adding high-to-low-resolution subnetwork structures. Then, it parallelly connects multi-resolution subnetworks and estimates keypoints on the high-resolution feature map outputted by the network. These methods utilize deep learning techniques to accurately detect the location of human joints in images, providing important information for subsequent 3D human pose recovery and providing the detected 2D joint points as input to a two-stage approach, the accuracy of 2D joint detection greatly benefits the two-stage method, and the network parameters are also lower compared to directly recovering end-to-end from the original image. Martinez et al. [22] constructed a method to recover 3D human pose from 2D joint points using simple linear layers and ReLU activation functions, utilizing high-dimensional modeling based on 2D keypoints to recover 3D human body poses, achieving promising results given the hardware limitations at the time. However, due to the simplicity of the network architecture, it could not fully capture the intricacies of the keypoints. Pavllo et al. [16] improved performance by using 2D joint point sequences as input and utilizing backprojection as a semisupervised method when labeled data are scarce. While this method provides a solution for datasets lacking 3D information annotations, the lack of precision in the recovery results can lead to cumulative errors, ultimately compromising the network’s accuracy.

Although many current methods have made various improvements to the input during network training, employing both original image inputs and 2D keypoints coordinates, and some have adopted semi-supervised approaches to address the challenge of acquiring 3D annotation data, there still lacks a method to fully utilize existing recovery data, leading to underutilization of data resources. Furthermore, most of these approaches feature simple structures and overlook the relationships between human body keypoints, resulting in poses that may not conform to conventional standards.

## 3. Methods

To extract spatial–temporal features and learn effective spatial–temporal correlations, a novel 3D pose estimation architecture with DASTFormer and BVLoss is proposed in this paper. Its outline is shown in Figure 2. Given an original video, the 2D pose extractor is first utilized to provide the 2D pose $P_{2D}$ as input. Then they perform linear embedding, temporal embedding, and spatial embedding through LTS embedding module. The output is marked as $LTS(P_{2D})$ and becomes the input of DASTFormer. During the DASTFormer encoding process, the relationship between spatial–temporal features and the 3D pose will be enhanced to improve prediction accuracy. Unlike conventional training routines, after completing a parameter update, BVloss will be introduced to conduct a second update based on data from the same batch. This will be beneficial for finding the global optimum of the proposed algorithm. More details are illustrated in the following subsection.

### 3.1. Dual Adaptive Spatial–Temporal Former

The DASTFormer architecture features a well-crafted three-branch system, integrating the spatial transformer block (STB) and temporal transformer block (TTB). Within this framework, the STB and TTB model different attributes through various dimensions. The STB calculates self-attention between joints based on spatial position embedding, learning the relationships between body joints. Meanwhile, the TTB calculates self-attention between joint frames based on temporal position embedding, learning the motion trajectories of the joints. Furthermore, DASTFormer includes modeling of the overall spatiotemporal dimension, accomplishing the modeling of spatiotemporal collaborative information through the mutual influence of the three branches. As illustrated in Figure 3, each branch seamlessly incorporates residual connections tailored for spatial or temporal transformer encoders, ensuring the efficacy of information propagation across the network. The thoughtful integration of these spatial–temporal branches serves to augment the model’s capacity to capture intricate spatial and temporal dependencies, thereby enhancing its overall performance. This multibranch design facilitates comprehensive feature extraction and context-aware learning, contributing to the robustness of the proposed DASTFormer architecture.

To fortify the connectivity between these blocks, two distinct modes, namely AtA and PuA, are employed, referencing the formulation in Equation (Equation 1). PuA uses the features of different branches predicted by the network to fit the weights of different branches, respectively. This method is stable globally, but directly assigning weights to branches can lead to ignoring the internal situation of the branches. The AtA is intended for remodeling the internal situation of the branches, which is achieved by calculating the internal attention, thereby obtaining an internal attention weight matrix. However, although AtA can distribute weights internally, it is susceptible to the influence of local errors. If AtA is fully adopted, it will lead to model instability. Considering comprehensively, the attention mechanism of the low-level features in the first few layers of the model has a relatively small impact. It is better at capturing local features and short-distance dependency relationships in the input sequence. Therefore, the AtA feature fusion method is adopted to better integrate local features. In the higher-level features of the model, as the number of layers increases, the attention mechanism can learn higher-level abstract representations. It understands the input data from a global perspective, and the PuA fusion method is adopted to make the model more stable globally.
(1)$$I_{i}=\left\{\begin{array}{c} LTS(P_{2D}),i=1\\ AtA\left(F_{S}^{i-1},F_{T}^{i-1},F_{ST}^{i-1}\right),1<i\leq 3\\ PuA\left(F_{S}^{i-1},F_{T}^{i-1},F_{ST}^{i-1}\right),3<i\leq N \end{array}\right.$$

In Equation (Equation 1), $I_{i}$ represents the input of the *i*-th block. $F_{S}^{i}$, $F_{T}^{i}$, $F_{ST}^{i}$, respectively, represent the spatial, temporal, and spatial–temporal features of the *i*-th block. AtA(a,b,c) and PuA(a,b,c) indicate attention-adaptive calculation and pure-adaptive calculation, respectively. Their forward propagation rules are shown in Figure 3, and their formula is defined in Equations (Equation 2)–(Equation 5).

According to Figure 3, the first two blocks adopt the AtA mode for spatial–temporal encoding, and the last three blocks use the PuA mode to perform three-branch fusion. The calculation of AtA is demonstrated in Equation (Equation 2),
(2)$$AtA\left(F_{S}^{i},F_{T}^{i},F_{ST}^{i}\right)=atMap^{i}\cdot apMap^{i}$$
where $atMap^{i}$ represents the self-attention map with $F_{S}^{i}$ as query, $F_{T}^{i}$ as key, and $F_{ST}^{i}$ as value. $apMap^{i}$ denotes the pure-adaptive output and is also seen as the output of $PuA(F_{S}^{i},F_{T}^{i},F_{ST}^{i})$.
(3)$$atMap^{i}=SF\left(\frac{F_{S}^{i}{\left(F_{T}^{i}\right)}^{T}}{\sqrt{d_{T}}}\right)F_{ST}^{i}$$
(4)$$apMap^{i}=Add\left(\alpha_{S}^{i}F_{S}^{i},\alpha_{ST}^{i}F_{ST}^{i},\alpha_{T}^{i}F_{T}^{i}\right)$$
where SF represents the Softmax operation, and scaling factor *d* represents the dimension of $F_{T}^{i}$. α denotes the adaptive weight from Equation (Equation 5). Add signifies the element-wise addition.
(5)$$\left[\alpha_{S}^{i},\alpha_{ST}^{i},\alpha_{T}^{i}\right]=SF\left(\omega \left[Cat\left(F_{S}^{i},F_{ST}^{i},F_{T}^{i}\right)\right]\right)$$
where Cat represents the concatenation of the values from the three branches along the last dimension, and ω denotes the dimension reduction of the last dimension to three dimensions.

Although AtA is meaningful, for efficiency, it is used only in the first two blocks. The last three blocks adopt the PuA mode. To verify its ability, the adaptive weights in the third block are visualized in Figure 4.

Upon scrutinizing the actual weights in the initial row, it becomes apparent that different keypoints exert diverse influences on 3D pose estimation, contingent on whether they are subjected to spatial, temporal, or spatial–temporal encoding. The postnormalization of these weights reveals a notable emphasis on the $F_{ST}$ branch within DASTFormer.

This observation suggests that the PuA mode adeptly assigns weights, discerning the spatial–temporal collaborative and independent impacts on 3D pose estimation. The deliberate design of the PuA mode plays a pivotal role in automatically adjusting these weights, underscoring its efficacy in uncovering intricate spatial–temporal dependencies. This nuanced approach enhances the model’s ability to discern the nuanced interplay between spatial and temporal features, contributing to the nuanced and context-aware 3D pose estimation facilitated by DASTFormer.


### 3.2. Supervised Training with Batch Variance

To refine the model’s spatial–temporal understanding, a two-round supervised training regimen is implemented on the same batch, as depicted in Figure 2. Following the initial training phase, where the proposed network derives the 3D pose $P^{First}3D$, the LTS module and DASTFormer undergo updates based on the joint information from $P^{First}3D$ and $P_{2D}$. This dual-input strategy leverages the spatial and temporal features encoded in $P_{2D}$ and the refined 3D pose $P_{3D}^{First}$ to iteratively enhance the model’s understanding of spatial–temporal relationships. The relevant calculations are summarized in Equation (Equation 6),
(6)$$L_{FST}=L_{3D}+\lambda_{T}L_{T}+\lambda_{2d}\sum L_{2d}^{\phi },\phi \in \left\{xy,xz,yz\right\}$$
where $L_{3D}$ is the distance error between ground truth and predicted pose $P_{3D}^{First}$. $L_{T}$ represents the velocity loss, as described in previous work [16]. And $L_{2d}^{\phi }$ denotes the keypoint projection loss between $P_{3D}^{First}$ and ground truth on *xy*, *xz*, and *yz* planes.
(7)$$L_{2d}^{\phi }=\sum_{t=1}^{T}\sum_{j=1}^{J}\left|\right|\hat{P_{t,j}^{\phi }}-P_{t,j}^{\phi }{\left|\right|}_{2},\phi \in \left\{xy,xz,yz\right\}$$
where *T* represents the frame length of $P_{2D}$, *J* represents the keypoint number, and $P_{t,j}^{xy}$ represents the projected position of *j*-th keypoint on the xy plane at frame *t*-th frame. Similarly, *xz* and *yz* planes.

After the first parameter update, $P_{2D}$ is once again employed to derive the 3D pose for the second round of supervised training, denoted as $P_{3D}^{Second}$. Subsequently, the batch variance loss (BVLoss) is applied for the second update on LTS and DASTFormer. During the secondary training process, we perform two updates for the same batch of data in each epoch operation. This means that during the second update, it is not only influenced by the original batch data but also by the first network prediction results. Since the overall trend of the network during training is to fit in a better direction, the prediction results after each parameter update are often better than before the update. Therefore, after sending the same batch through the network twice, the first prediction can be used as a negative sample. This means that the process of updating network parameters is influenced by the data equivalent to two batches, which appropriately alleviates the limitation of the graphics card batch size with the graphics memory size remaining unchanged. This iterative training approach enhances the model’s capacity to capture intricate spatial–temporal dependencies, leading to improved overall performance. The formulation of BVLoss is provided in Equation (Equation 8).
(8)$$L_{BV}=Max\left(L_{FST}^{Cur}-L_{FST}^{Old}+m,0\right)$$
where $L_{FST}^{Old}$ represents the loss between $P_{2D}$ and $P_{3D}^{First}$, and $L_{FST}^{Cur}$ indicates the loss between $P_{2D}$ and $P_{3D}^{Second}$, *m* is a hyperparameter obtained from experience. So far, the final objective function can be summarized as Equation (Equation 9),
(9)$$L_{Ob}=\gamma L_{BV}+L_{FST}^{Cur}$$
where the γ is the weight of BVLoss. We detail an example of the codes in Algorithm 1.

**Algorithm 1:** Pseudo-code of training process
**Input**: $I_{\mathrm{img}}$**Parameter**: 2D pose extractor $T_{2D}$, DASTFormer $T_{\mathrm{DAST}}$, $T_{s}$, $T_{t}$ and $T_{st}$ represents the spatial, temporal, spatial-temporal block, respectivelyInitialization $T_{2D}$ from posetrack**while**  *in train stage*  **do**(  Extract $P_{2D}$ = $T_{2D}\left(I_{\mathrm{img}}\right)$  **for**  *t in time*  **do**(    Achieve LTS Embedding of 2D Keypoints $F_{2D}$ according to Equation (1)    **for**  *i in depth*  **do**      Extract $F_{S}$=$T_{s}\left(F_{2D}\right)$, $F_{T}$=$T_{t}\left(F_{2D}\right)$, $F_{ST}$=$T_{st}\left(F_{2D}\right)$      Calc. different branches of weights $\alpha_{S}$, $\alpha_{T}$, $\alpha_{ST}$ according to Equation (5)      Calc. the output of PuA and is also seen as the $apmap^{i}$ according to Equation (4)      **if**  *i in AtAList*  **then**        Calc. the self-attention map $atmap^{i}$ according to Equation (3)        Calc. the output result of AtA according to Equation (2)      **end**      Achieve final output according to Equation (1) (    **end**    **if**  *t in first update*  **then**      Optimize $T_{\mathrm{DAST}}$ according to Equations (6) and (7) // (
first update    **else**      Optimize $T_{\mathrm{DAST}}$ according to Equations (6)–(8)
// (
second update    **end**  **end**
**end**



## 4. Experiments

### 4.1. Datasets and Implementation Details

We trained and tested on two mainstream datasets (human3.6M [56], humaneva [57]), and the evaluation indicators are MPJPE and P-MPJPE, which are transformed by rotation and alignment before MPJPE. They are used to measure the average error between the estimated 3D joint point positions and the true joint point positions. Specifically, for each joint point, the Euclidean distance between the estimated joint point and the true joint point is computed, and then the distance is averaged over all joint points. P-MPJPE is often used to evaluate the performance of attitude estimation algorithms in a more comprehensive way. P-MPJPE inherits the concepts of MPJPE while introducing the idea of Procrustes analysis to more robustly measure the error in attitude estimation results. For frame f and skeleton S, the detailed calculation formula is as follows:(10)$$E_{MPJPE}\left(f,S\right)=\frac{1}{N_{S}}\sum_{i=1}^{Ns}∥m_{f,S}^{\left(f\right)}\left(i\right)-m_{\mathrm{gt},S}^{\left(f\right)}\left(i\right){)∥}_{2}$$
where $N_{S}$ represents the number of joints contained in the skeleton S, and taking the average value of MPJPE for the sequence. $m_{f,S}^{\left(f\right)}\left(i\right)$ is the pose estimator f that returns the coordinates of the i-th joint point of skeleton *S* at frame *f*.

Human3.6M is a large-scale public dataset for 3D human pose estimation research, which is currently the most important dataset based on 3D human pose research. The dataset consists of 3.6 million 3D human poses and corresponding images, captured by four calibrated high-resolution 50 HZ cameras and precise 3D joint positions and angles from high-speed motion capture systems. Each actor was also subjected to 3D laser scanning to ensure accurate capture. The BMI index range of these action actors is between 17 and 29, ensuring a moderate range of body shape variability and different activity levels. The subjects wore their own daily clothing, rather than special motion capture suits, to maintain a sense of realism as much as possible. Data from seven subjects were used for training and validation, while data from the other four subjects were used for testing. The data were organized into 15 training actions, including various asymmetric walking poses (such as walking with hands in pockets, walking with a shoulder bag), sitting poses, lying poses, various waiting poses, and other types of poses. The actors were given detailed tasks with examples to help them plan a set of stable poses between repetitions to create training, validation, and testing sets. Then, during the execution of these tasks, the actors had considerable freedom to go beyond a strict interpretation of the task. We adopted a 17-joint 3D skeleton, following previous works [9,49,58]. Training used S1, S5, S6, S7, S8, with evaluation on S9 and S11.

HumanEva includes six actions by four actors, e.g., Walking, Jogging, Boxing, and Greeting. It established the quantitative evaluation of human pose estimation using well-defined metrics in 2D and 3D. We used actions by S1, S2, S3 for training and reserved the remaining actions (Walking, Jogging) for testing.

In the experiments, we configured the video frames to a length of 243 for the Human3.6 dataset. The network was trained for a total of 80 epochs, using a batch size of 10. The model depth was fixed at 5, and we employed eight multihead self-attention mechanisms. Both the first and second updates were executed with a learning rate of 0.0002. The specific parameters are shown in Table 1.

### 4.2. Comparison with the State-of-the-Art

**Results on Human3.6M.** In our experiments, we generated two-dimensional joint data using the method proposed in [16]. Table 2 displays the results, comparing MPJPE and P-MPJPE for 15 actions using our approach versus other methods. To ensure a fair comparison, we maintained consistent input sequence lengths (in this case, 243 frames) and kept other modules unchanged. Noticeably, categories such as phone, pose, smoke, and walk exhibit significant improvements when using the BVLoss model compared to the model without it. This suggests that through iterative parameter updates on the same dataset, the network can effectively learn the positional variations of global information, enabling secondary fusion and ultimately enhancing the model’s performance.

The model without BVLoss outperforms most state-of-the-art methods in various categories. This effectively validates our approach of incorporating both temporal and spatial positional information into DASTFormer. The designed attention-adaptive and pure-adaptive modes successfully facilitate global positional feature fusion. Our method achieved MPJPE of 39.6 mm for Protocol 1 and P-MPJPE of 33.4 mm for Protocol 2, surpassing P-STMO [21] by 3.2 mm in terms of MPJPE (7.5%).

Furthermore, we conducted a comprehensive comparison of our method with those utilizing ground truth data, as detailed in Table 3. The results underscore the significant superiority of our approach, surpassing all other methods and achieving a notable 2.2 mm improvement in terms of MPJPE (11.6%) over Diffpose [59].

**Table 2 sensors-24-04422-t002:** Pose estimation results under Protocol 1 and Protocol 2 on the Human3.6M Dataset. (**Top table**) Results for MPJPE under Protocol 1. (**Bottom table**) Results for P-MPJPE under Protocol 2; *T* denotes the number of input frames estimated by the respective approaches, (*) indicates the transformer-based methods. The best and second-best results are highlighted in bold and underlined formats, respectively. A lower Avg metric is preferable.

Protocol #1	Publication	T	Dir1.	Disc.	Eat	Greet	Phone	Photo	Pose	Pur.	Sit	SitD.	Smoke	Wait	WalkD.	Walk	WalkT.	*Avg*
LCN [60]	ICCV 2019	1	46.8	52.3	44.7	50.4	52.9	68.9	49.6	46.4	60.2	78.9	51.2	50.0	54.8	40.4	43.3	52.7
Xu et al. [61]	CVPR 2021	1	45.2	49.9	47.5	50.9	54.9	66.1	48.5	46.3	59.7	71.5	51.4	48.6	53.9	39.9	44.1	51.9
Liu et al. [58]	CVPR 2020	243	41.8	44.8	41.1	44.9	47.4	54.1	43.4	42.2	56.2	63.6	45.3	43.5	45.3	31.3	32.2	45.1
Chen et al. [9]	TCSVT 2021	81	42.1	43.8	41.0	43.8	46.1	53.5	42.4	43.1	53.9	60.5	45.7	42.1	46.2	32.2	33.8	44.6
Wehrbein et al. [62]	ICCV 2021	200	38.5	42.5	39.9	41.7	46.5	51.6	39.9	40.8	**49.5**	**56.8**	45.3	46.4	46.8	37.8	40.4	44.3
MHFormer [43](*)	CVPR 2022	351	39.2	43.1	40.1	40.9	44.9	51.2	40.6	41.3	53.5	60.3	43.7	41.1	43.8	29.8	30.6	43.0
P-STMO [21](*)	ECCV 2022	243	38.9	42.7	40.4	41.1	45.6	**49.7**	40.9	39.9	55.5	**5**9.4	44.9	42.2	42.7	29.4	29.4	42.8
STCFormer-L [29](*)	CVPR 2023	243	38.4	41.2	**36.8**	38.0	42.7	50.5	38.7	38.2	52.5	56.8	41.8	38.4	40.2	26.2	27.7	40.5
Ours (wo BVLoss,*)		243	36.8	40.2	39.4	34.4	42.2	50.7	37.8	36.8	51.9	60.0	42.1	38.5	37.9	26.0	26.5	40.0
Ours (w BVLoss,*)		243	**36.8**	**39.7**	39.3	**34.3**	**40.9**	50.6	**36.8**	**36.7**	50.9	59.0	**41.4**	**38.4**	**37.9**	**25.3**	**25.8**	**39.6**
ine **Protocol #2**	**Publication**	**T**	**Dir1.**	**Disc.**	**Eat**	**Greet**	**Phone**	**Photo**	**Pose**	**Pur.**	**Sit**	**SitD.**	**Smoke**	**Wait**	**WalkD.**	**Walk**	**WalkT.**	* **Avg** *
Liu et al. [58]	CVPR 2020	243	32.3	35.2	33.3	35.8	35.9	41.5	33.2	32.7	44.6	50.9	37.0	32.4	37.0	25.2	27.2	35.6
PoseFormer [20](*)	ICCV 2021	81	34.1	36.1	34.4	37.2	36.4	42.2	34.4	33.6	45.0	52.5	37.4	33.8	37.8	25.6	27.3	36.5
Chen et al. [9]	TCSVT 2021	81	33.1	35.3	33.4	35.9	36.1	41.7	32.8	33.3	42.6	49.4	37.0	32.7	36.5	25.5	27.9	35.6
MHFormer [43](*)	CVPR 2022	351	31.5	34.9	32.8	33.6	35.3	39.6	32.0	32.2	43.5	48.7	36.4	32.6	34.3	23.9	25.1	34.4
P-STMO [21](*)	ECCV 2022	243	31.3	35.2	32.9	33.9	35.4	**39.3**	32.5	31.5	44.6	**48.2**	36.3	32.9	34.4	23.8	23.9	34.4
GLA-GCN [63](*)	ICCV 2023	243	32.4	35.3	**32.6**	34.2	35.0	42.1	32.1	31.9	45.5	49.5	36.1	32.4	35.6	23.5	24.7	34.8
Ours (wo BVLoss,*)		243	31.3	33.8	33.9	29.6	34.6	39.7	31.1	32.2	44.4	51.1	36.5	31.4	32.9	22.3	23.0	33.8
Ours (w BVLoss,*)		243	**31.1**	**33.7**	33.8	**29.4**	**34.0**	39.6	**30.3**	**31.4**	**43.5**	49.7	**36.0**	**31.3**	**32.8**	**22.0**	**22.6**	**33.4**

**Results on HumanEVA.** To assess the generalizability of our model, we evaluated our approach on the HumanEVA dataset. Following the methodology introduced in [28], we used the MixSTE method for data preprocessing. Utilizing 43 frames of 2D pose sequences as input to the model, we adapted the sequence length due to the dataset’s limited samples and shorter sequences compared to the Human3.6M dataset. Furthermore, we used a smaller data sample stride (interval = 1). As depicted in Table 4, our method consistently achieved the best performance in terms of MPJPE. Furthermore, comparing models with and without BVLoss demonstrated a significant improvement in all categories, confirming the improved generalizability of our model. We achieved an MPJPE of 9.6mm with ground truth data, showcasing the superior pose accuracy of our approach.

### 4.3. Ablation Study

To assess the impact of each component and design in the proposed model, we conducted ablation experiments on the Human3.6M dataset under Protocol 1 with MPJPE evaluation.

**Impact of Temporal–Spatial Branch.** For video-based human pose estimation tasks, achieving accurate results heavily relies on the effective interaction of temporal–spatial features. As illustrated in Table 5, we initially present the results of the structural design of the temporal-spatial blocks within DASTFormer. In this context, S−T and T−S correspond to the sequential configurations of the temporal–spatial blocks, with S−T exhibiting superior performance in sequential mode. This highlights the significance of the sequential structure in capturing temporal dependencies and enhancing the overall accuracy of the model. S+T represents a design of parallel structures, and considering the superiority of S−T in the sequential structure, we introduced a three-branch structure S+S−T+T. It is evident that, when the model employs the same pure adaptive mode, the three-branch network structure exhibits superior performance. This underscores the effectiveness of our method in capturing nuanced inter-frame global temporal–spatial dependencies. The intricate design of the three-branch structure allows for enhanced adaptability and improved representation of complex temporal–spatial relationships in video-based human pose estimation tasks. From the results of DASTFormer with a batch size of 24, it can be observed that there is a minor improvement in the outcomes with the increase in batch size. Furthermore, even when the batch size is the same at 24, our final structure still outperforms the best-performing structure among the aforementioned four, which is the S+S−T+T configuration. Under an input sequence length of 243 frames, we achieved a 1.3 mm improvement compared to S+T, indicating the effectiveness of the proposed method in capturing larger inter-frame global temporal–spatial dependencies.

**Impact of BVLoss.** At the bottom of Table 5, we validated the results of adding BVLoss to DASTFormer. The experiments indicate that the inclusion of BVLoss slightly improves its performance during one more training. In addition, we investigated the impact of different batch sizes on the results. Our observations indicate that the inclusion of BVLoss in the S+S−T+T branch, along with a batch size set to 24, resulted in performance inferior to the accuracy achieved by our DASTFormer with a batch size of 10. This comparison underscores the efficiency of our model in achieving high accuracy with a reduced number of data. The effectiveness of our designed BVLoss is evident in its ability to enhance learning from the available data, thereby contributing to improved model performance. This outcome validates the rationale behind incorporating BVLoss and highlights its role in reinforcing the model’s capacity to learn and adapt to diverse training scenarios.

### 4.4. Visualization and Analysis

For visual analysis, we conducted a visualization on the Human3.6M dataset, as depicted in Figure 5, comparing the pose estimation results to the ground truth 3D poses. The visual representation highlights the superior accuracy of our method over PoseFormer. Furthermore, we emphasize the robust performance of our approach across diverse scenarios, reinforcing its effectiveness in practical applications.

In addition, we also visualize on videos outside the dataset, as shown in Figure 6. The green arrows in groups a and b indicate accurate pose estimation, while the red arrows in group a signify deviations in the estimated pose. This illustrates the strong generalization capability of our model, demonstrating excellent performance. Even in challenging scenarios such as joint overlap or occlusion, the estimated poses maintain a high level of consistency with the actual human body poses.

## 5. Conclusions and Discussion

In summary, 3D human pose estimation holds paramount importance in numerous domains, serving as a foundational technology for advancements in areas such as virtual and augmented reality, rescue based on remote sensing, and sports science. Due to the lack of depth information in 2D inputs, spatial and temporal cues play an important role in inferring 3D poses. Therefore, this work proposed a novel architecture with DASTFormer and one-more supervised training. The DASTFormer can consider the spatial–temporal cooperative and independent effects on 3D pose inference by two adaptive learning. The one-more supervised training with batch variance loss is different from the common training strategy. It can conduct a two-round parameter update on the same batch data, which will not only better explore the potential relationship between spatial–temporal encoding and 3D poses, but also alleviate the batch-size limitation of graphics cards on transformer-based frameworks. To validate the effectiveness of the proposed method, a large number of experiments were tested on Human3.6 and HumanEVA datasets. The experimental results demonstrate that DASTFormer and one-more supervised training with BVLoss can significantly improve the MPJPE and P-MPJPE.

Of course, it still needs more researchers to work together to facilitate the development of 3D human pose estimation in actual application scenarios. In real scenes, human actions are often affected by occlusion and changes in perspective, which lead to challenges to action recognition. However, video-based 3D pose estimation can obtain pose information from multiple perspectives, allowing for a more comprehensive understanding of human movements and mitigating the effects of occlusion and perspective changes. In addition, the results of 3D pose estimation can also serve as auxiliary input for action recognition tasks. By combining them with pixel-level clues such as parsing and segmentation, more representative and discriminative features can be extracted to improve action recognition. Using the results of 3D pose estimation, human motions can be edited and optimized more accurately. By mapping the estimated joint position to the 3D human model, the pose and motion will be adjusted. To further improve the quality of motions, it can also be combined with physical simulation technology to make the action of the generated model more in line with the laws and constraints of the real world. In addition, it can also be used in combination with motion capture technology when obtaining posture from real human body data and applying it to generate motions, so as to achieve highly realistic animation effects.

We envision that continued research in this direction will not only foster advancements in 3D human pose estimation but also contribute to broader fields such as action recognition, animation, human–computer interaction, and virtual reality. In the future, exploring the intersection of the aforementioned directions will ultimately enrich various applications and societal benefits.

## Figures and Tables

**Figure 1 sensors-24-04422-f001:**
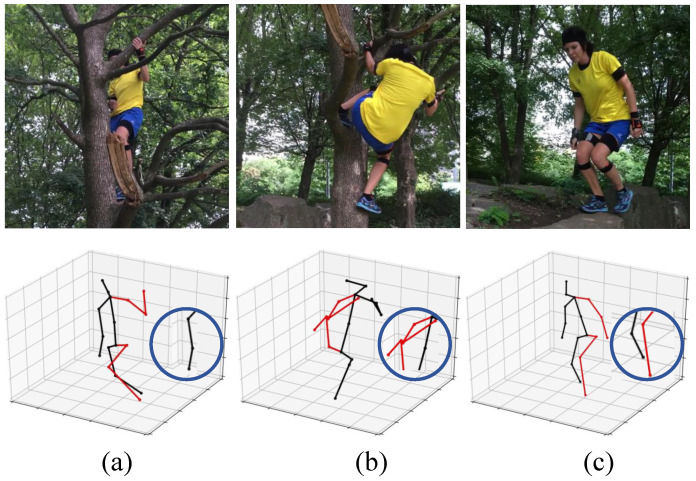
Some failed visualize examples of 3D pose estimation using a single image from the wild dataset 3DPW [19] as input. The first line refers to the raw inputs.The second line shows the 3D pose estimated by videoposed3D [16]. (**a**) When the body is obstructed, the estimated pose of the right arm deviates. (**b**) When the body is in a complex posture, there is unexpected overlap in the 3D pose of upper body. (**c**) When the background is cluttered, there is an incorrect association between the left and right legs.

**Figure 2 sensors-24-04422-f002:**
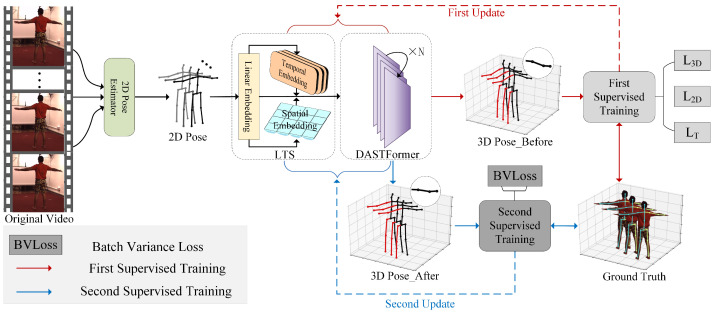
Outline of the proposed method. LTS and DASTFormer are responsible for feature encoding. BVLoss is only applied for the second supervised training and guides the 3D Pose_After results surpass the 3D Pose_Before. Best viewed in color.

**Figure 3 sensors-24-04422-f003:**
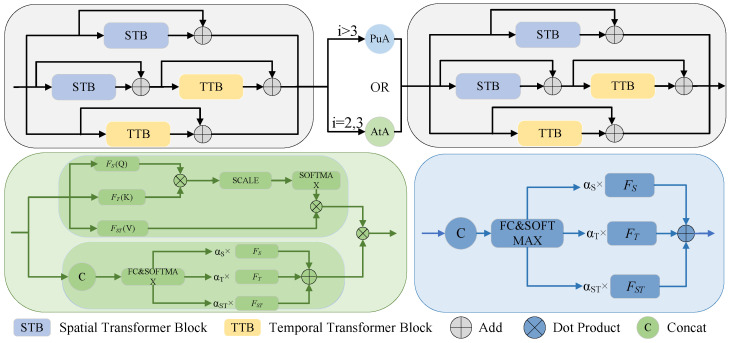
DASTFormer. DASTFormer consists of *N* spatial–temporal blocks (in grey) with two adaptive modes. The green subgraph on the left represents the attention-adaptive mode (AtA), while the blue part on the right shows the pure-adaptive mode (PuA).

**Figure 4 sensors-24-04422-f004:**
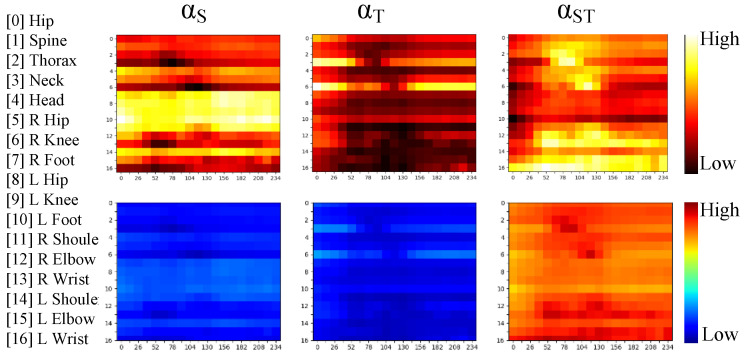
Visualizations of PuA in Block 3. The first row presents the real weights, while the second row depicts the normalized weights. Each column represents the attention weights $\alpha_{S},\alpha_{T}$, and $\alpha_{ST}$, respectively. The *x*-axis and *y*-axis represent frame number and keypoint id.

**Figure 5 sensors-24-04422-f005:**
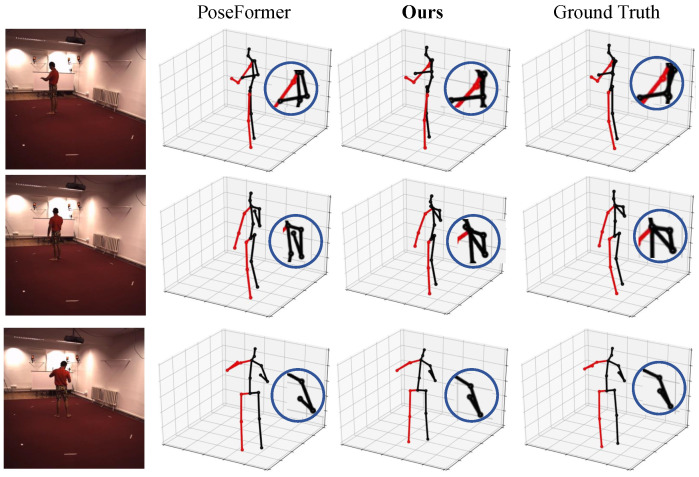
Qualitative comparison with PoseFormer [20] and GT. Our method is qualitatively compared with PoseFormer [20] on some actions in Human3.6M. The blue circles highlight positions where our method achieves superior results.

**Figure 6 sensors-24-04422-f006:**
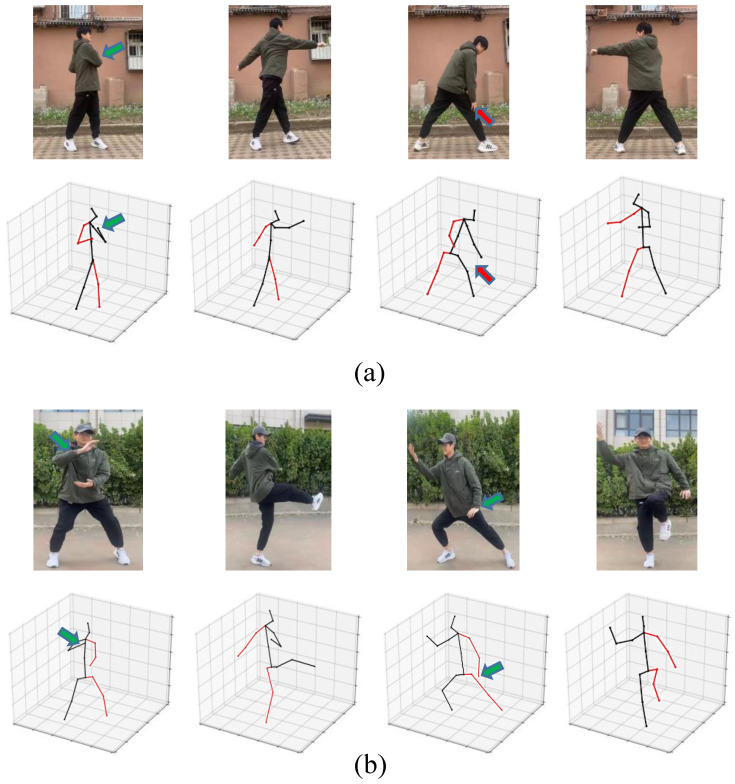
Visualization under challenging in real-world videos. The green arrows indicate accurate pose estimation, while the red arrows signify deviations in the estimated pose. The labels (**a**) and (**b**) represent two different videos.

**Table 1 sensors-24-04422-t001:** Human Posture Estimation with Parameter Settings for Different Datasets in Training Phase.

	config	Human3.6M	HumanEVA
optimizer	learning_rate	0.00002	
	weight_decay	0.01	
	lr_decay	0.99	
model	maxlen	243	43
	dim_feat	256	
	mlp_ratio	4	
	depth	5	
	dim_rep	512	
	num_heads	8	
data	data_stride	81	22
	num_joints	17	15
	Batchsize	10	64

**Table 3 sensors-24-04422-t003:** Pose estimation results of MPJPE on Human3.6M under Protocol 1 using 2D ground truth keypoints as input. The best results are highlighted in bold.

Protocol #1	Publication	T	Dir1.	Disc.	Eat	Greet	Phone	Photo	Pose	Pur.	Sit	SitD.	Smoke	Wait	WalkD.	Walk	WalkT.	*Avg*
Liu et al. [58]	CVPR 2020	243	34.5	37.1	33.6	34.2	32.9	37.1	39.6	35.8	40.7	41.4	33.0	33.8	33.0	26.6	26.9	34.7
PoseFormer [20]	ICCV 2021	81	30.0	33.6	29.9	31.0	30.2	33.3	34.8	31.4	37.8	38.6	31.7	31.5	29.0	23.3	23.1	31.3
MHFormer [43]	CVPR 2022	351	27.7	32.1	29.1	28.9	30.0	33.9	33.0	31.2	37.0	39.3	33.0	31.0	29.4	22.2	23.0	30.5
P-STMO [21]	ECCV 2022	243	28.5	30.1	28.6	27.9	29.8	33.2	31.3	27.8	36.0	37.4	29.7	29.5	28.1	21.0	21.0	29.3
Diffpose [59]	CVPR 2023	243	18.6	19.3	18.0	18.4	18.3	21.5	21.5	19.1	23.6	22.3	18.6	18.8	18.3	12.8	13.9	18.9
Ours (w BVLoss)		243	**16.8**	**17.8**	**16.5**	**15.7**	**16.7**	**17.8**	**18.4**	**18.9**	**20.9**	**21.0**	**17.3**	**15.3**	**15.5**	**10.2**	**10.8**	**16.7**

**Table 4 sensors-24-04422-t004:** The MPJPE on HumanEva testset under Protocol 1. The best result is highlighted in bold.

Protocol #1	Publication	T		Walk			Jog		*Avg*
Pavllo et al. [16]	CVPR2019	81	14.0	12.5	27.1	20.3	17.9	17.5	18.2
zheng et al. [20]	ICCV2021	43	14.4	10.2	46.6	22.7	13.4	13.4	20.1
MixSTE [28]	CVPR2022	43	12.7	10.9	17.6	22.6	15.8	17.0	16.1
Ours (w/o BVLoss)		43	10.3	6.8	14.4	14.6	8.5	8.9	10.6
Ours (w BVLoss)		43	**9.6**	**5.7**	**12.3**	**13.8**	**7.9**	**8.1**	**9.6**

**Table 5 sensors-24-04422-t005:** An ablation analysis of individual components within our methodology, evaluated using MPJPE in millimeters on the Human3.6M dataset.

Component	Batchsize	PuA	AtA	BVLoss	MPJPE
T−S	10	-	-	✗	41.1
S−T	10	-	-	✗	40.9
S+T	10	✓	✗	✗	41.7
S+S−T+T	10	✓	✗	✗	40.4
DASTFormer	10	✓	✓	✗	40.0
DASTFormer	10	✓	✓	✓	39.6
S+S−T+T	24	✓	✗	✓	40.0
DASTFormer	24	✓	✓	✓	39.5

## Data Availability

Data are contained within the article.

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
