# Peer review of "Learning Temporal–Spatial Contextual Adaptation for Three-Dimensional Human Pose Estimation"

_sensors, 2024, doi:10.3390/s24134422_

Round 1
Reviewer 1 Report
Comments and Suggestions for Authors
The idea is not completely novel, and the overall paper should be improved in that the logic behind the method design should be further explained.
Comments on the Quality of English LanguageThe manuscript needs extensive revision. Some errors:
1. Existing excellent methods primarily focus on "explore"
2. In addition, "an" one-more supervised training
Reviewer 2 Report
Comments and Suggestions for Authors
In this work, the authors propose a double adaptive space-time front (DASTFormer). It realizes spatiotemporal co-coding through two modes: Attention Adaptive (AtA) and Pure Adaptive (PuA). This design can enhance the different effects of temporal and spatial coding on each keypoint, allowing for more accurate estimation of 3D poses. On this basis, a single-supervised training strategy with batch variance loss (BVLoss) was designed to seek the global optimum. However, there are some problems needed to be solved before considered for publication.
Comment I. Abstract: This paper introduces a model called Dual Adaptive Spatio-Temporal Transformer (DASTFormer). The model achieves spatio-temporal collaborative encoding through two modes: Attentional-Adaptive (AtA) and Pure-Adaptive (PuA). It is recommended to mention this abbreviation in the abstract for easier readability by the readers.
Comment II. The overview of the proposed method in Figure 2. It is recommended to redraw the image of the LTS module as the text in it is not clear.
Comment III. In Section 3. Methods, it is not detailed how the "attention-adaptive" and "pure-adaptive" modes interact in the "dual-adaptive spatial-temporal former". It is suggested to further explain how the combination of spatial and temporal information enhances the accuracy of 3D pose estimation.
Comment IV. This work mentions the issue regarding the bulk size limit of the graphics card, but does not elaborate on how to fix it. More information on how to overcome hardware limitation may be helpful to readers. And the results of the experiment at a batch size of 24 are missing in the "Impact of BVLoss" experiment.
Comment V. While experiments were conducted on the Human3.6M and HumanEVA datasets as mentioned in Section 4.1, there is a lack of validation in real-world application scenarios. The performance in real-world scenarios might be more convincing.
Comment VI. Some relevant references are missing, and citations are recommended. E.g.:
1. Besides, Chaiet al. [22] combines global adaptation and local generalization in PoseDA, which is a simple and effective unsupervised domain adaptation framework for 3D human pose estimation. [R1]
[R1]Joint-bone Fusion Graph Convolutional Network for Semi-supervised Skeleton Action Recognition. IEEE Transactions on Multimedia, vol.25, pp.1819-1831, 2023.
2. Incorporating Long Short-Term Memory (LSTM) units with normalized layers and temporal smoothness constraints for training. [R2,R3]
[R2]Skeleton-based action recogni tion with spatial reasoning and temporal stack learning. In Proc. European Conference on Computer Vision (ECCV), 2018: 106-121.
[R3]Skeleton-based action recognition using LSTM and CNN. In Proc. IEEE International Conference on Multimedia & Expo Workshops (ICMEW), 2017: 585-590.
Comments on the Quality of English LanguageThe English can be improved.
Reviewer 3 Report
Comments and Suggestions for Authors
Too much of the text.
Some technical improvements are needed.
